# Lifestyle Related Cancer Risk and Protective Behaviors Vary among a Convenient Sample of Physically Active, Young-to-Middle-Aged Adults 18–49

**DOI:** 10.3390/ijerph20136305

**Published:** 2023-07-06

**Authors:** Aldenise P. Ewing, Gregory C. Chang, Abhishek V. Henry, Jordyn A. Brown, Mahmood A. Alalwan, Donte T. Boyd, Daniel Marshall, Skylar McElwain, Alicia L. Best, Claudia F. Parvanta, Bruce L. Levin, Cathy D. Meade, Clement K. Gwede

**Affiliations:** 1College of Public Health, The Ohio State University, Columbus, OH 43210, USA; chang.1582@buckeyemail.osu.edu (G.C.C.); henry.1164@buckeyemail.osu.edu (A.V.H.); alalwan.2@buckeyemail.osu.edu (M.A.A.); marshall.1227@buckeyemail.osu.edu (D.M.); mcelwain.35@osu.edu (S.M.); 2Gillings School of Global Public Health, The University of North Carolina, Chapel Hill, NC 27599, USA; jordyn.brown@unc.edu; 3College of Social Work, The Ohio State University, Columbus, OH 43210, USA; boyd.465@osu.edu; 4College of Public Health, University of South Florida, Tampa, FL 33612, USA; abest@usf.edu (A.L.B.); cparvanta@usf.edu (C.F.P.); levin@usf.edu (B.L.L.); 5Department of Health Outcomes and Behavior, Moffitt Cancer Center, Tampa, FL 33612, USA; cathy.meade@moffitt.org (C.D.M.); clement.gwede@moffitt.org (C.K.G.)

**Keywords:** chronic disease prevention, recreational sports, young-to-middle-aged adults, risk behaviors, protective behaviors

## Abstract

It is an assumption that physically active adults lead an overall healthy lifestyle. To examine this assumption, we administered a cross-sectional, web-based survey to a sample of young-to-middle-aged US adults between 18 and 49 who self-reported participation in at least one recreational sporting event in the past month. Logistic regressions were conducted to examine demographic characteristics associated with cancer risk and protective behaviors. Gender was represented equally (N = 938), and the average age was 32 years (SD: 8.4). Most participants reported >three days of moderate- to high-intensity physical activity (79%), but not meeting fruit and vegetable consumption guidelines (78%). Many reported current tobacco use (32%), binge drinking at least once in the past 30 days (62%), and suboptimal sun protection use (67%). Participation in lifestyle-related cancer risk and protective behaviors varied based on age, sex, education, routine doctor visits, perceived overall health, health-information-seeking behavior (how participants obtained health information), or team-based sport participation in regression models. Future interventions should be tailored to address varied cancer risk profiles among even physically active adults to encourage multiple healthy behavior changes.

## 1. Introduction

Current recommendations suggest that adults should perform at least 150 min of moderate-intensity exercise or 75 min of vigorous exercise per week for optimal health benefits [1]. Americans who meet these physical activity guidelines are 21–29% less likely to be diagnosed with coronary heart disease and 10–20% less likely to be diagnosed with cancer than those who do not meet the guidelines [2,3]. Despite the proven health benefits, however, it is reported that less than a third of US adults meet the physical activity recommendations [4]. 

Recreational sports, defined as competitive physical games, such as basketball or baseball, are played for fun as opposed to professionally and provide one of many opportunities for individuals to engage in physical activity [5]. The recreational sporting environment gathers wide attention from young and middle-aged adults and provides opportunities to encourage healthy behaviors for both short and long-term benefits. Nonetheless, it is an untested assumption that physically active adults lead healthier lifestyles than physically inactive individuals and are more attuned to their long-term health, as few studies provide complete health profiles for physically active adults. 

Complete lifestyle behavioral health profiles of physically active adults could facilitate research on the combined effects of healthy (i.e., physical activity, fruit and vegetable consumption, etc.) and unhealthy (i.e., tobacco use, alcohol consumption, etc.) behaviors [6]. Furthermore, limited research exists across the lifespan of physically active adults that examines lifestyle behaviors associated with an increased or decreased risk of developing chronic diseases known to occur later in life [7,8,9,10,11,12,13]. Within that context, this study was designed to identify specific lifestyle behaviors (e.g., fruit and vegetable consumption, days of physical activity, current tobacco use, alcohol consumption, and use of sun protection) known to be associated with cancer and subsequent participant profiles. We hypothesized that a higher proportion of physically active adults would report engagement in healthy lifestyle behaviors (e.g., fruit and vegetable consumption, days of physical activity, and use of sun protection) than unhealthy lifestyle behaviors (e.g., current tobacco use and alcohol consumption). 

## 2. Materials and Methods

A cross-sectional, web-based survey was administered between January and April 2019 on social media sites (e.g., Facebook, Instagram, and Twitter) via Qualtrics. The principal investigator partnered with several organizations that hosted recreational sporting events to facilitate recruitment among members of special interest social media group pages. Before beginning the survey, participants reviewed informed consent and voluntarily agreed to participate. Participants were asked a series of questions to ascertain engagement in cancer-related risk and protective behaviors. In exchange for their time, participants received a USD 10 e-gift card for a commercial retailer. Further information on the survey design can be found elsewhere [14,15]. 

### 2.1. Sample 

Inclusion criteria were adults 18–49 years of age, participation in at least one recreational sporting event per month, and no prior cancer diagnosis. Non-probability sampling techniques (e.g., volunteer, self-selection, and nonproportional quota sampling) were used for recruitment [16]. 

### 2.2. Predictor Variables

Predictor variables included age (continuous), sex, race, ethnicity, marital status, sexual orientation, education, employment, income, healthcare coverage, routine doctor visits, cancer screening history, perceived overall health, health-information-seeking behavior (how participants obtain health information), and team-based sport participation. Open-ended text responses allowed participants to report their participation in different types of recreational sports. Individual-based recreational sports consisted of activities such as boxing, jogging, weightlifting, cycling, martial arts, swimming, rock climbing, and yoga. Team-based sports consisted of softball, volleyball, bowling, soccer, recreational sports clubs, flag football, basketball, etc.

### 2.3. Self-Reported Lifestyle Behaviors

Lifestyle-related cancer risk and protective behaviors were assessed using items adapted from the Health Information National Trends Survey (HINTS) and the Behavioral Risk Factor Surveillance System (BRFSS) questionnaire to allow comparisons to the general population of the same age in the US. The HINTS and BRFSS are large US national surveys that contain validated questions examining chronic-disease-related risk factors [17,18]. Items adapted from HINTS 5 Cycle 3 (2019) assessed physical activity, fruit and vegetable consumption, current cigarette smoking, and current e-cigarette and “vape” use. Items assessing binge drinking by sex and the use of sun protection were directly adopted from the BRFSS (2018). According to the BRFSS, binge drinking is classified as five or more drinks on one occasion for men and four or more for women. A single item that assessed chewing tobacco, snuff, or snus use was directly adopted from HINTS-FDA Cycle 2 (2017) questionnaire. Overall, five dichotomous chronic disease-related risk and protective behaviors were identified: physical activity, fruit and vegetable consumption, current tobacco use (i.e., cigarette use, e-cigarette or vape use, chew, snuff, or snus use), binge drinking at least once in the past 30 days, and use of sun protection. 

In our analysis, respondents were categorized as meeting physical activity guidelines if they reported engagement in moderate- to high-intensity physical activity or exercise, such as brisk walking, bicycling at a regular pace, swimming at a regular pace, or heavy gardening, three or more times a week. Meeting fruit and vegetable recommendations was defined as reporting fruit consumption of at least “1 to 2 cups” and vegetable consumption of at least “2 to 3 cups” daily. Respondents who reported being current cigarette smokers, e-cigarette or vape users, or users of chewing tobacco, snuff, or snus were categorized as current tobacco users. Men who consumed at least five or more alcoholic beverages and women who consumed four or more alcoholic beverages on one or more occasions in the past 30 days were categorized as those who participated in binge drinking. Optimal use of sun protection was met if respondents reported always wearing or wearing sunscreen most of the time when they were outside on a sunny day for one hour or more. 

### 2.4. Data Analysis

Descriptive statistics consisted of counts and frequencies for the sociodemographic characteristics included in our analysis. We conducted univariate and multiple logistic regressions to examine predictors for each of the lifestyle-related risk and protective behaviors. Statistical significance was determined at the alpha 0.05 level. We performed a complete case analysis for the final analytic sample included in the models. All analyses were conducted in SAS v.9.4 (SAS, Inc., Cary, NC, USA).

## 3. Results

### 3.1. Demographics

The analytic sample included 938 respondents who reported participation in one or more recreational sporting events per month, aged 18 to 49 years old, and did not have a prior cancer diagnosis (Table 1). The mean age of the participants was 32 years (SD = 8.4) and nearly half (49.6%) identified as female sex at birth. Most reported non-Hispanic ethnicity (90.6%) and either White (42.4%) or Black/African American (49.2%) race. A little more than a third of participants reported attending some college or receiving a 2-year degree (37.1%), full-time employment (65.1%), and earning between USD 30,000 and 59,999 (36.5%). Most participants reported having health care coverage (87.8%), a routine doctor visit within the last two years (79.5%), and no history of past cancer screening (70%). More than half the sample reported being single (58.9%) and most reported heterosexual orientation (87.8%). Many participants perceived themselves to have good (50.1%) or average (25.9%) overall health. More than half of participants reported seeking health information from a person or healthcare provider (53.5%) followed by digital or online resources (39.9%). Lastly, more than two-thirds of respondents reported participation in individual-based sports (64%). Reported ethnicity, education, health care coverage, income, marital status, information-seeking behavior, and sexual orientation were missing for ~24% of participants. “Missing” in Table 1 and Table 2 refers to the number of participants who did not respond to an item. Missing data were not included in the total percentages.

Counts and proportions for each lifestyle-related risk and protective behavior are reported in Appendix A. Univariate analyses of predictors and each lifestyle-related risk and protective behavior performed are reported in Table 2.

Physical activity. Sex, race, healthcare coverage, cancer screening history, perceived overall health, and team-based sport participation were significantly associated with reporting three or more days of physical activity in the univariate regression models (*p*-value < 0.05). Most of the sample reported at least three or more physically active days (78.6%) with a significantly higher proportion reported among men compared to women (81.5% vs. 75.6%), those with health care compared to no health care (80.6% vs. 70.9%), and those who participated in team-based sports (Table 2). Those who reported White race (82.4%) had the highest proportion of being physically active for three or more days a week followed by those who self-reported some Other race (78.98%), and then Black race (74.1%) (Table 2). Additionally, a higher proportion of men (10.4%) compared to women (6.0%) reported physical activity as the only healthy behavior performed (Appendix A). Among those physically active, the highest proportion reported a combination of two (31.2%) or three healthy behaviors (31.2%). Among those who performed all possible healthy behaviors alongside physical activity, a higher proportion were women (5.4% vs 3.6% of men). In contrast, a comparable proportion of physically inactive women (11.3%) and men (10.5%) reported participation in all other unhealthy behaviors (Appendix A).

Fruit and vegetable consumption. Although most participants (77.9%) did not meet recommendations for fruit and vegetable consumption, education, cancer screening history, and perceived overall health were independently significantly associated with meeting fruit and vegetable recommendations (*p* < 0.05). Respondents with a professional, graduate, or doctorate degree (28.6%) had the highest proportion of meeting fruit and vegetable recommendations, and those with less than a high school diploma or with only a high school diploma reported the lowest proportion (12.3%). Higher proportions of respondents meeting fruit and vegetable recommendations were seen among those who reported a cancer screening history compared to those never screened (28.8% vs. 19.2%) and those who reported excellent overall health (33.5%) (Table 2). 

Current tobacco use (i.e., cigarette use, e-cigarette or vape use, chew, snuff, or snus use). Sex, education, healthcare coverage, perceived overall health, information-seeking behavior, routine doctor visits, and team-based sport participation were significantly associated with current tobacco use (31.7%) in the univariate regression models (*p* < 0.05). Higher proportions of current tobacco use were seen in men compared to women (38.8% vs. 24.5%), those with no health care coverage compared to respondents with health care coverage (39.1% vs. 28.6%), and those who only participated in individual recreational sports compared to team-based recreational sports (39.2% vs. 19.9%). Those with a high school diploma or less than a high school diploma (42.8%) had higher proportions of tobacco use compared to those with a professional degree (15.3%). Participants who reported excellent overall health had a lower proportion of current tobacco use (27.2%) compared to those who reported poor/terrible health (44.7%) (Table 2). 

Binge drinking at least once in the past 30 days. Binge drinking in the past 30 days was reported by 62.1%. Only employment was significantly associated with reported binge drinking (*p* < 0.05). Participants who were employed full-time had the highest proportion of binge drinking (63.1%) compared to those who reported part-time employment or other (53.6% and 52.5%, respectively) (Table 2). 

Use of sun protection. Regular use of sun protection was reported by 33.4% of the total sample. Education, cancer screening history, perceived overall health, age, and routine doctor visits were significantly associated with the use of sun protection (*p* < 0.05). Higher proportions of regular use of sun protection were reported among those who had routine doctor visits compared to no routine doctor visits (35.7% vs. 24.3%), and those with a cancer screening history compared to no cancer screening history (40.3% vs. 30.3%). Participants reported as professionals, graduates, or with a doctorate had the highest proportion of regular sun protection (41.2%). Those with some college or two-year degree and those with a high school diploma or less reported similar proportions of sun protection use (28.3% and 28.0%). Respondents who reported excellent overall health (45.6%) had the highest proportion of regular use of sun protection compared to those who reported terrible/poor overall health (15.8%) (Table 2).

### 3.2. Multiple Logistic Regression Models

Results of multiple regression models between each lifestyle-related risk and protective behavior and sociodemographic characteristic are reported in Table 3. 

Physical activity. Self-reported sex at birth, age, and perceived overall health were associated with engagement in moderate-intensity physical activity at least three days a week. Men versus women (adjusted odds ratio (aOR): 1.86; 95% CI: 1.18, 2.89) and participants who reported excellent (aOR: 3.78; 95% CI: 1.86, 7.70) or good (aOR: 1.89; 95% CI: 1.19, 2.99) versus average perceptions of their overall health had higher odds of achieving at least three days a week of moderate-intensity physical activity. Older participants demonstrated lower odds of achieving at least three or more days a week of moderate-intensity physical activity (Age, aOR: 0.97; 95% CI: 0.94, 1.00). 

Fruit and vegetable consumption. Participants reporting excellent (aOR: 2.88; 95% CI: 1.59, 5.23) vs. average perceptions of their overall health had higher odds of meeting recommendations for fruit and vegetable consumption. 

Current tobacco use. Men had 1.58 times the odds of reporting current tobacco use compared to women (95% CI: 1.06, 2.34). Participants with less than a high school diploma or high school diploma had 1.98 times the odds of being current tobacco users compared to those with a four-year degree (95% CI: 1.15, 3.43). Assessing perceived overall health, participants who reported having excellent health vs. average health had 0.42 times the odds of being current tobacco users (95% CI: 0.23, 0.75). Participants who reported good (aOR: 0.62; 95% CI: 0.40–0.96) or excellent (aOR: 0.42; 95% CI: 0.23–0.75) vs. average perceptions of overall health had lower odds of reporting current tobacco use. Looking at information-seeking behaviors, participants who sought information through printed or other resources versus persons or providers had 2.6 times the odds of being current tobacco users (95% CI: 1.30, 5.19). Participants who engaged in team-based sports had 0.37 times the odds of being current tobacco users compared to those who did not participate in team-based sports (95% CI: 0.24, 0.57). 

Binge drinking in the past 30 days. There were no significant differences in self-reported binge drinking in the past 30 days for this sample. 

Use of sun protection. Perceived overall health and routine doctor visits were associated with regular use of sun protection. Participants who reported having excellent vs. average perceived overall health had 2.3 times the odds of regularly using sun protection (95% CI: 1.35, 3.92). Participants who self-reported having their most recent doctor’s visit over two years ago had 0.51 times the odds compared to those who had a doctor’s visit within the past 1 or 2 years (95% CI: 0.30, 0.85).

## 4. Discussion

Our study was designed to provide a more complete lifestyle behavioral health profile and test the assumption that physically active individuals lead overall healthy lifestyles with participation in other healthy behaviors. Although most of our sample of young-to-middle-aged adults 18 to 49 who reported participation in recreational sports met physical activity guidelines, they unexpectedly reported more frequent participation in unhealthy behaviors including not meeting fruit and vegetable consumption guidelines, binge drinking, and suboptimal sun protection use. Well over half of the physically active adults reported no current tobacco use. Based on comparisons to publicly available data for US adults of the same age, our sample demonstrated comparable or higher levels of days of physical activity, fruit and vegetable consumption, and use of sun protection [19]. Thus, an asset-based approach, leveraging physical activity, may further reduce cancer-related risk attributed to diet and UV exposure among this unique subgroup. Antithetical to what was assumed by their self-reported participation in recreational sports and days of physical activity, this sample may face a greater risk for developing cancer due to more frequent use of tobacco and alcohol binge drinking than the general population of the same age [20]. 

Our results indicated one-third of physically active adults engaged in current tobacco use and two-thirds reported binge drinking behavior. Previously published research on adolescents and young adults corroborate our findings of reported alcohol and drug use among sport participants [21,22]. With trends showing alarming increases in the use of tobacco and alcohol among adolescents and young adults, there are missed opportunities to promote health communication campaigns and interventions for reducing cancer risks earlier in life [20]. By identifying demographic characteristics and lifestyle behaviors associated with health benefits and risks in our study, professionals may build on this and develop targeted interventions to prevent or control alcohol and drug use behaviors from youth into early adulthood. Particularly, our study suggests that leveraging socioenvironmental attributes, such as team-based sports participation, may be ideal for smoking cessation or reduced-alcohol-consumption-centered messaging for this subgroup of young-to-middle-aged adults.

Based on our findings of associated sex, education, and information-seeking behavior among current tobacco users in our sample, more anti-tobacco use campaigns should highlight the negative effects in gender-specific, low-literacy, print-based educational materials to meet the self-reported characteristics of our sample. Consumption of tobacco in any form inhibits optimal lung functioning, particularly for recreational sport participation, and physically active adults may stand to benefit from more health information that exposes the negative effects of tobacco use on sport-related performance. There are reportedly few existing evidence-based cessation resources available for adults with the lowest levels of education [23]. Although research provides evidence on the effect of health literacy levels on perceptions of smoking health risk, future research may consider testing effective messages with recreational sport participants to ensure optimal reach. 

Future interventions to address varied cancer risk profiles among physically active adults should encourage the adoption of multiple healthy behavior changes such as healthy diet + tobacco cessation, sun protection use + tobacco cessation, or healthy diet + tobacco cessation + reduced/no alcohol consumption. Findings from our present study may be used for tailoring future interventions to promote multiple healthy behavior changes among an otherwise “unworried and well” group of young-to-middle-aged adults who participate in recreational sporting events. Based on some of our findings, encouraging adults to increase physical activity may serve as a catalyst for adopting at least one other healthy behavior (e.g., no tobacco use, sun protection, fruit and vegetable consumption). Future research should further examine this hypothesis that physical activity may be leveraged in an asset-based approach to promote other healthy behaviors among recreational sport participants. 

### 4.1. Limitations 

Limitations of this study include self-reporting and the use of a convenient sample of physically active, young-to-middle-aged adults; thus, findings are not generalizable to young-to-middle-aged adults in the US. Selection bias is also a possibility as health-conscious or physically active adults might have been more likely and willing to complete the survey. Additionally, not all recreational sports involve the same level of fitness and those varied levels of intensity of recreational activities may have had an unidentified effect on decisions to engage in healthy or unhealthy behaviors. Further investigation by sport type may lead to a better understanding of which activities support overall healthy lifestyles rather than an umbrella of all recreational sports. Missing data may also be a concern for the following predictors: ethnicity, education, health care coverage, income, marital status, information-seeking behavior, and sexual orientation. Missing data for these predictors could influence response bias in the sample if the data were not missing at random. Lastly, although the focus of this study was to highlight associated behaviors with recreational sport participation, there is inherently no way to determine which behavior (i.e., recreational sport participation or participation in risky behaviors) occurred first due to the temporality of cross-sectional surveys.

### 4.2. Strengths 

By adopting items from national surveys, our study creates opportunities for general population comparisons. These comparisons are important for identifying potential areas in which physically active, young-to-middle-aged adults aged 18–49 may be more at risk for developing chronic diseases than others of the same age, thus highlighting the need for further research and public health intervention. Along with identifying risky behaviors, this study highlights distinct sociodemographics of potentially higher-risk young-to-middle-aged adults. By highlighting both health-benefitting and risky behaviors, future intervention research may consider an asset-based approach that builds upon healthy habits—or perceptions—while discouraging unhealthy habits among physically active, young-to-middle-aged adults aged 18–49. 

Due to the exploratory nature of this study, future research should assess outcomes for chronic disease-specific conditions and participant profiles (i.e., ethnicity, sexual orientation, sport type, etc.). Future research should include additional items for measurement of risk behaviors as participants may have selected the more socially acceptable and desirable responses for each of the single items assessing risk behaviors (i.e., binge drinking).

## 5. Conclusions

Cancer is a leading cause of death worldwide. The initiation of prevention behaviors early in adulthood is one strategy for reducing lifetime chronic disease risk and death. Existing data depict a growing trend in participation in sports and exercise among adults of all ages [12]. However, until now, no other study has explored cancer-related risk or protective behaviors among physically active adults aged 18–49. As adults find alternative ways to engage in physical activity through sport, it is important that participants, as well as public health professionals, are cognizant of health risks and health-enhancing behaviors that may accompany recreational sport participation. By focusing on this young-to-middle-aged adult population, earlier interventions may perhaps more effectively reduce both short- and long-term chronic-disease-related outcomes among adults.

## Figures and Tables

**Table 1 ijerph-20-06305-t001:** Characteristics of a sample of physically active adults ages 18–49 (N = 938).

Characteristic	Total	
	n	%
Age (Mean, SD)	31.61 (8.39)	
Sex		
Male	473	50.43
Female	465	49.57
Race		
White	365	42.44
Black or African American	423	49.19
Other	72	8.37
Missing	78	
Ethnicity		
Non-Hispanic	649	90.64
Hispanic	67	9.36
Missing	222	
Education		
Less than High School or High School Diploma	138	19.38
Some college or 2-year degree	264	37.08
4-year degree	191	26.83
Professional, Graduate or Doctorate	119	16.71
Missing	226	
Employment		
Employed, Full Time	464	65.17
Employed, Part Time	85	11.94
Other	163	22.89
Missing	226	
Income		
<29 k	198	29.12
30–59 k	248	36.47
60–99 k (ref)	154	22.65
>100 k	80	11.76
Missing	258	
Health Care Coverage		
Yes	625	87.78
No	87	12.22
Missing	226	
Routine Doctor Visits		
Yes	672	79.53
No	173	20.47
Missing	93	
Cancer Screening History		
Yes	278	29.99
No	649	70.01
Missing	11	
Marital Status		
Married	225	31.42
Single	422	58.94
Other	69	9.64
Missing	222	
Sexual Orientation		
Heterosexual	619	87.8
Gay or Lesbian	41	5.82
Bisexual	33	4.68
Other	12	1.7
Missing	233	
Perceived Overall Health		
Excellent	158	19.32
Good	410	50.12
Average	212	25.92
Poor/Terrible	32	4.65
Missing	120	
Information Seeking Behavior		
Person/Provider	381	53.51
Digital/Online	284	39.89
Print/Other	47	6.6
Missing	226	
Team-Based Sport Participation		
Team-Based	338	36.03
Individual	600	63.97

**Table 2 ijerph-20-06305-t002:** Univariate analyses of predictors and lifestyle behaviors performed among a sample of physically active adults ages 18–49.

Characteristic	Lifestyle Behaviors									
	Met Days of Physical Activity (≥3)		Met Fruit and Vegetable Recommendation		Current Tobacco Use		Binge Drinking		Sun Protection Use	
	**n (%)**	***p* Value** †	**n (%)**	***p* Value** †	**n (%)**	***p* Value** †	**n (%)**	***p* Value** †	**n (%)**	***p* Value** †
Sex		0.0339		0.9343		<0.0001		0.9035		0.5112
Overall	682 (78.57)		193 (22.11)		274 (31.67)		521 (62.10)		282 (33.37)	
Male	357 (81.51)		98 (22.22)		169 (38.76)		273 (61.90)		139 (32.33)	
Female	325 (75.58)		95 (21.99)		105 (24.48)		248 (62.31)		143 (34.46)	
Race		0.022		0.4756		0.4304		0.9366		0.7153
Overall	644 (78.06)		184 (22.22)		265 (32.16)		498 (62.25)		272 (33.37)	
White	291 (82.44)		79 (22.25)		111 (31.44)		219 (62.93)		112 (32.09)	
Black or African American	297 (74.06)		93 (23.19)		135 (33.83)		238 (61.82)		137 (34.77)	
Other	56 (78.87)		12 (16.67)		19 (26.39)		41 (61.19)		23 (31.94)	
Ethnicity		0.6789		0.6598		0.2736		0.4046		0.6041
Overall	565 (79.24)		154 (21.51)		214 (29.97)		410 (59.42)		234 (32.68)	
Non-Hispanic	514 (79.44)		141 (21.73)		190 (29.37)		367 (58.91)		214 (32.97)	
Hispanic	51 (77.27)		13 (19.40)		24 (35.82)		43 (64.18)		20 (29.85)	
Education		0.2467		0.0109		<0.0001		0.2185		0.0245
Overall	563 (79.41)		153 (21.49)		212 (29.86)		408 (59.46)		233 (32.72)	
Less than High School or High School Diploma	103 (74.64)		17 (12.32)		59 (42.75)		79 (58.96)		39 (28.26)	
Some college or 2-year degree	211 (79.92)		55 (20.83)		89 (33.84)		146 (57.03)		74 (28.03)	
4-year degree	148 (78.72)		47 (24.61)		46 (24.08)		121 (65.76)		71 (37.17)	
Professional, Graduate or Doctorate	101 (84.87)		34 (28.57)		18 (15.25)		62 (55.36)		49 (41.18)	
Employment		0.0906		0.1913		0.595		0.0348		0.3789
Overall	563 (79.41)		153 (21.49)		212 (29.86)		408 (59.48)		233 (32.72)	
Employed, Full Time	370 (80.09)		102 (21.98)		133 (28.79)		280 (63.06)		158 (34.05)	
Employed, Part Time	60 (70.59)		12 (14.12)		29 (34.12)		45 (53.57)		29 (34.12)	
Other	133 (82.10)		39 (23.93)		50 (30.67)		83 (52.53)		46 (28.22)	
Income		0.3153		0.2524		0.6029		0.5435		0.0678
Overall	534 (78.88)		148 (21.76)		205 (30.26)		400 (60.79)		226 (33.24)	
<29 k	148 (75.13)		34 (17.17)		63 (31.82)		111 (57.81)		66 (33.33)	
30–59 k	195 (78.95)		61 (24.60)		76 (30.77)		151 (63.71)		70 (28.23)	
60–99 k	124 (80.52)		33 (21.43)		40 (25.97)		89 (58.55)		55 (35.71)	
>100 k	67 (84.81)		20 (25.00)		26 (32.91)		49 (63.64)		35 (43.75)	
Healthcare Coverage		0.0398		0.4535		0.0462		0.7138		0.5471
Overall	563 (79.41)		153 (21.49)		212 (29.86)		408 (59.48)		233 (32.72)	
Yes	502 (80.58)		137 (21.92)		178 (28.57)		359 (59.73)		207 (33.12)	
No	61 (70.93)		16 (18.39)		34 (39.08)		49 (57.65)		26 (29.89)	
Routine Doctor Visits		0.174		0.6389		0.0134		0.7043		0.0048
Overall	658 (78.24)		187 (22.13)		270 (32.03)		504 (61.84)		282 (33.38)	
Yes	530 (79.22)		151 (22.47)		201 (30.00)		401 (62.17)		240 (35.71)	
No	128 (74.42)		36 (20.81)		69 (39.88)		103 (60.59)		42 (24.28)	
Cancer Screening History		0.0418		0.0018		0.7903		0.5206		0.0047
Overall	682 (78.57%)		193 (22.10%)		274 (31.68%)		521 (62.10%)		282 (33.37%)	
Yes	218 (82.89)		76 (28.79)		81 (31.03)		160 (63.75)		104 (40.31)	
No	464 (76.69)		117 (19.21)		193 (31.95)		361 (61.39)		178 (30.32)	
Marital Status		0.2929		0.7622		0.5669		0.9457		0.1391
Overall	565 (79.24)		154 (21.51)		214 (29.97)		410 (59.42)		234 (32.68)	
Married	180 (80.36)		52 (23.11)		69 (30.80)		129 (58.90)		85 (37.78)	
Single	326 (77.62)		87 (20.62)		121 (28.74)		240 (59.41)		127 (30.09)	
Other	59 (85.51)		15 (21.74)		24 (34.78)		41 (61.19)		22 (31.88)	
Sexual Orientation		0.7769		0.7559		0.5758		0.2332		0.155
Overall	559 (79.63)		153 (21.70)		208 (29.59)		404 (59.41)		231 (32.77)	
Heterosexual	492 (79.87)		134 (21.65)		180 (29.17)		350 (58.72)		195 (31.50)	
Gay or Lesbian	33 (80.49)		7 (17.07)		16 (39.02)		28 (71.79)		20 (48.78)	
Bisexual	24 (72.73)		9 (27.27)		9 (27.27)		21 (63.64)		12 (36.36)	
Other	10 (83.33)		3 (25.00)		3 (25.00)		5 (41.67)		4 (33.33)	
Perceived Overall Health		<0.0001		0.0001		0.0061		0.992		<0.0001
Overall	636 (78.13)		179 (21.88)		259 (31.74)		485 (61.47)		271 (33.13)	
Excellent	140 (88.61)		53 (33.54)		43 (27.22)		96 (61.94)		72 (45.57)	
Good	335 (82.11)		91 (22.20)		116 (28.29)		242 (61.11)		142 (34.63)	
Average	139 (66.19)		30 (14.15)		83 (39.52)		126 (62.07)		51 (24.06)	
Poor/Terrible	22 (57.89)		5 (13.16)		17 (44.74)		21 (60.00)		6 (15.79)	
Information Seeking Behavior		0.0724		0.5115		0.0008		0.91		0.2559
Overall	563 (79.41)		153 (21.49)		212 (29.86)		408 (59.48)		233 (32.72)	
Person/Provider	311 (81.63)		88 (23.10)		108 (28.42)		219 (60.16)		135 (35.43)	
Digital/Online	221 (78.37)		55 (19.37)		78 (27.56)		162 (58.91)		84 (29.58)	
Print/Other	31 (67.39)		10 (21.28)		26 (55.32)		27 (57.45)		14 (29.79)	
Team-Based Sport Participation		0.0388		0.7735		<0.0001		0.2226		0.1081
Overall	682 (78.57)		193 (22.12)		274 (31.68)		521 (62.10)		282 (33.37)	
Team-Based	277 (82.20)		73 (21.60)		67 (19.88)		191 (59.50)		102 (30.18)	
Individual	405 (76.27)		120 (22.43)		207 (39.20)		330 (63.71)		180 (35.50)	

† *p*-value for chi-squared test comducted using complete case analysis for predictor and outcome variables.

**Table 3 ijerph-20-06305-t003:** Adjusted odds ratio from multiple models assessing predictors of lifestyle-related cancer risk and protective behaviors.

Characteristics	Physical Activity ‡			Fruit and Vegetable Consumption ‡			Current Tobacco Use ‡			Binge Drinking in the Past 30 Days ‡			Use of Sun Protection ‡		
	aOR	95% CI	*p*-Value	aOR	95% CI	*p*-Value	aOR	95% CI	*p*-Value	aOR	95% CI	*p*-Value	aOR	95% CI	*p*-Value
Age (Mean, SD)	0.967	(0.94–0.996)	0.0254	1.007	(0.979–1.036)	0.6339	0.996	(0.97–1.023)	0.7756	0.988	(0.965–1.012)	0.3389	1.006	(0.982–1.031)	0.6102
Sex			0.0073			0.3585			0.0233			0.7782			0.2872
Male vs. Female	1.845	(1.18–2.885)		0.824	(0.545–1.246)		1.578	(1.064–2.339)		0.951	(0.668–1.352)		0.819	(0.567–1.183)	
Race			0.0746			0.6986			0.3396			0.6981			0.9812
Other vs. Black or African American	1.048	(0.488–2.251)		0.791	(0.367–1.704)		1.442	(0.697–2.982)		1.328	(0.687–2.568)		0.949	(0.484–1.858)	
White vs. Black or African American	1.667	(1.056–2.633)		0.85	(0.559–1.293)		1.322	(0.886–1.973)		1.07	(0.748–1.53)		1.012	(0.697–1.47)	
Ethnicity			0.2014			0.8014			0.4245			0.6355			0.4563
Hispanic vs. Non-Hispanic	0.641	(0.324–1.268)		0.914	(0.454–1.839)		1.288	(0.692–2.397)		1.15	(0.645–2.053)		0.787	(0.418–1.479)	
Education			0.5911			0.0971			0.0157			0.4617			0.3594
Less than High School or High School Diploma vs. 4-year degree	1.062	(0.576–1.958)		0.444	(0.224–0.879)		1.984	(1.148–3.431)		0.915	(0.543–1.543)		0.793	(0.461–1.366)	
Professional, Graduate or Doctorate vs. 4-year degree	1.526	(0.773–3.011)		1.077	(0.608–1.907)		0.65	(0.335–1.26)		0.725	(0.43–1.222)		0.965	(0.573–1.624)	
Some college or 2-year degree vs. 4-year degree	1.263	(0.749–2.13)		0.895	(0.551–1.454)		1.348	(0.841–2.16)		0.75	(0.49–1.148)		0.681	(0.438–1.058)	
Employment			0.2426			0.1326			0.9241			0.1181			0.7303
Other vs. Full Time	1.072	(0.604–1.903)		1.455	(0.851–2.488)		1.026	(0.626–1.682)		0.646	(0.412–1.015)		0.825	(0.507–1.345)	
Part Time vs. Full Time	0.623	(0.335–1.159)		0.688	(0.335–1.417)		1.125	(0.628–2.017)		0.674	(0.392–1.159)		0.98	(0.554–1.733)	
Income			0.6115			0.2816			0.3506			0.559			0.2284
30–59 k vs. 60–99 k	1.13	(0.646–1.979)		1.35	(0.805–2.265)		1.044	(0.629–1.735)		1.275	(0.815–1.994)		0.72	(0.45–1.15)	
<29 k vs. 60–99 k	0.863	(0.454–1.637)		0.825	(0.43–1.585)		0.997	(0.555–1.791)		1.056	(0.624–1.789)		1.086	(0.625–1.887)	
>100 k vs. 60–99 k	1.407	(0.64–3.096)		0.948	(0.482–1.866)		1.751	(0.902–3.4)		1.393	(0.771–2.518)		1.122	(0.622–2.022)	
Health Care Coverage			0.1962			0.6339			0.7537			0.9637			0.8166
No vs. Yes	0.677	(0.374–1.224)		1.17	(0.613–2.232)		0.916	(0.531–1.581)		0.988	(0.59–1.656)		1.07	(0.605–1.89)	
Routine Doctor Visits			0.3331			0.6972			0.0921			0.3625			0.0099
Over two years vs. Within the past 1 or 2 years	0.773	(0.458–1.303)		0.898	(0.521–1.546)		1.501	(0.936–2.408)		0.818	(0.53–1.261)		0.508	(0.303–0.85)	
Marital Status			0.3466			0.9486			0.3056			0.5979			0.3056
Never Married vs. Married	1.021	(0.627–1.662)		0.939	(0.594–1.483)		0.829	(0.533–1.291)		1.202	(0.81–1.783)		0.754	(0.501–1.137)	
Other vs. Married	1.808	(0.792–4.132)		0.912	(0.453–1.834)		1.358	(0.713–2.584)		1.248	(0.685–2.274)		0.691	(0.368–1.297)	
Sexual Orientation			0.9167			0.3354			0.3461			0.4198			0.0772
Bisexual vs. Heterosexual	0.875	(0.369–2.077)		1.816	(0.779–4.233)		0.866	(0.358–2.099)		1.164	(0.533–2.542)		1.382	(0.62–3.079)	
Gay or Lesbian vs. Heterosexual	1.222	(0.517–2.886)		0.684	(0.285–1.639)		1.933	(0.928–4.03)		1.776	(0.85–3.709)		2.133	(1.086–4.19)	
Other vs. Heterosexual	1.462	(0.26–8.21)		1.919	(0.415–8.882)		0.911	(0.209–3.977)		0.688	(0.182–2.593)		2.732	(0.668–11.169)	
Perceived Overall Health			<0.0001			0.0007			0.0225			0.9002			0.0034
Excellent vs. average	3.78	(1.855–7.704)		2.88	(1.585–5.231)		0.419	(0.233–0.752)		0.877	(0.528–1.456)		2.303	(1.352–3.922)	
Good vs. average	1.887	(1.193–2.985)		1.412	(0.852–2.34)		0.62	(0.401–0.959)		1.007	(0.673–1.507)		1.485	(0.959–2.298)	
Poor/Terrible vs. average	0.462	(0.196–1.091)		0.432	(0.112–1.656)		0.931	(0.391–2.215)		1.167	(0.493–2.761)		0.484	(0.162–1.45)	
Information Seeking Behavior			0.1308			0.4061			0.0124			0.8533			0.3041
Digital/Online vs. Person/Provider	0.829	(0.538–1.277)		0.766	(0.507–1.158)		0.903	(0.611–1.333)		0.989	(0.699–1.4)		0.783	(0.543–1.128)	
Print/Other vs. Person/Provider	0.468	(0.222–0.991)		0.743	(0.327–1.687)		2.602	(1.304–5.19)		0.828	(0.427–1.604)		0.677	(0.328–1.395)	
Team-Based Sport Participation			0.44			0.7758						0.7153			0.1344
Team sport vs. Individual sport	1.197	(0.759–1.888)		0.941	(0.619–1.431)		0.372	(0.244–0.565)	<0.0001	0.935	(0.652–1.341)		0.749	(0.514–1.093)	

‡ Models adjusted for age, sex, race, ethnicity, education, health care coverage, income level, marital status, employment, perceived overall health, cancer screening history, health-information-seeking behavior, sexual orientation, routine doctor visit, and participation in team-based sports.

## Data Availability

The data presented in this study are available on request from the corresponding author. The data are not publicly available due to privacy and ethical concerns.

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
