# Peer review of "Lifestyle Related Cancer Risk and Protective Behaviors Vary among a Convenient Sample of Physically Active, Young-to-Middle-Aged Adults 18–49"

_ijerph, 2023, doi:10.3390/ijerph20136305_

Round 1

Reviewer 1 Report

Well written, with some minor errors.  What is snus?

Many limitations here due to the sample used - selection bias since health conscious people are more likely to volunteer for this type of study.  Reporting bias is also a big issue here due to self-report (make sure to mention this).

Conclusion is a bit rambling and needs clarity as to what conclusions are being drawn from these data.

Table 3 is needs better formatting - illegible as is.

Author Response

  1. Well written, with some minor errors.  What is snus?

Response: A variant of dry tobacco. Snus is powdered tobacco that may be used like chewing tobacco and is placed between the lips and gum.

  1. Many limitations here due to the sample used - selection bias since health-conscious people are more likely to volunteer for this type of study.  Reporting bias is also a big issue here due to self-report (make sure to mention this).

Response: We appreciate the reviewer’s comment and acknowledge limitations of our study. We describe our sample (section 2.1) and highlight collection of self-reported data (section 2.3) in the Limitations section of the manuscript (section 4.1).

Lastly, we have added the following statement within the limitations section (lines 429-430):

“Selection bias is also a possibility as health-conscious participants might have been more likely and willing to complete the survey.”

  1. Conclusion is a bit rambling and needs clarity as to what conclusions are being drawn from these data.

Response: We appreciate the reviewer’s comment and have thoroughly revised our discussion section to better reflect our main findings.

  1. Table 3 is needs better formatting - illegible as is.

Response: To increase legibility, we have added row/column borders and shading for tables 2 and 3.

Reviewer 2 Report

Suggestion:  Minor revision

1.       Is there a baseline/threshold for the term ‘binge drinking’? Does it need to be over a certain volume of alcohol in one seating? If so – please do outline it. While ‘binge’ is a term commonly used in our daily lives, it would e useful to know if it can be quantified, especially for a manuscript

2.       Please could you indicate what ‘missing’ means in your table 1 and 2. Could the data not be collected? And if there is a chunk of data missing for each category – how is the % adding up to nearly 100%? Please clarify.

3.       For table 2 – is it possible to present the data for each category may be as a pie chart with % instead of long (an complicated) tables. Currently – it is loaded with data but does not necessarily provide an insight. Having it shown in the form of a chart (pie seems most feasible to me – but it is up to you) will enhance the understanding of the readers visually

4.       Also – if the ‘missing’ data is not included in the % calculations, you can below each pie chart the number of data that was missing.

5.       Could you please re-arrange the layout of Table 3 (and the others). It is confusing to read and decipher the categories. Use rows and column if needed to indicate boundaries. Currently it feels as if one set of data is flowing  into the other

6.       Considering there is abundant evidence linking increasing age to cancer – did you find any such connection in your observation? While your tables and the demographics section of results have the ages mentioned, you haven’t discussed this factor in your discussion. It would definitely be advantageous to analyse your data with respect to increasing risk of cancer with ageing.

7.       You may expand your final conclusions and future directions to up to 2 paragraphs

8.       Author contributions are not outlined in the required section at the end of the manuscript

Minor editing of English language required

Author Response

  1. Is there a baseline/threshold for the term ‘binge drinking’? Does it need to be over a certain volume of alcohol in one seating? If so – please do outline it. While ‘binge’ is a term commonly used in our daily lives, it would be useful to know if it can be quantified, especially for a manuscript

Response: We now addressed adoption of survey items from US National surveys and define binge drinking.

Line 100-101: “Items assessing binge drinking by sex and use of sun protection were directly adopted from the BRFSS (2018)”

Line 101-102: “according to the BRFSS, binge drinking is defined by five or more drinks on one occasion for men and four or more for women.”

  1. Please could you indicate what ‘missing’ means in your table 1 and 2. Could the data not be collected? And if there is a chunk of data missing for each category – how is the % adding up to nearly 100%? Please clarify.

Response: We provide clarification to the “missing” data referenced in our Tables. Missing refers to the number of participants who did not provide a response for the specific questions. Missing data was not included in the total percentages and therefore, the percentage for the non-missing data is equal to 100%.” To add further clarification, we do mention complete case analysis performed in the final models.

  1. For table 2 – is it possible to present the data for each category may be as a pie chart with % instead of long (and complicated) tables. Currently – it is loaded with data but does not necessarily provide any insight. Having it shown in the form of a chart (pie seems most feasible to me – but it is up to you) will enhance the understanding of the readers visually. Also – if the ‘missing’ data is not included in the % calculations, you can below each pie chart the number of data that was missing.

Response: To increase legibility, we have added row/column borders and shading for tables 2 and 3.

  1. Could you please re-arrange the layout of Table 3 (and the others). It is confusing to read and decipher the categories. Use rows and column if needed to indicate boundaries. Currently it feels as if one set of data is flowing into the other

Response: To increase legibility, we have added row/column borders and shading for tables 2 and 3.

  1. Considering there is abundant evidence linking increasing age to cancer – did you find any such connection in your observation? While your tables and the demographics section of results have the ages mentioned, you haven’t discussed this factor in your discussion. It would definitely be advantageous to analyze your data with respect to increasing risk of cancer with ageing.

Response: We thank the reviewer for this comment to provide clarification. Our study was designed to provide a complete health profile and test the assumption that physically active individuals lead healthier lifestyles and are more attuned to their long-term health needs than physically inactive individuals through survey of young-to-middle aged adults 18 to 49 who reported participation in recreational sports.

Although history of cancer was not our outcome of interest, we do agree that age is the leading risk factor for cancer diagnoses. There has been an abundance of evidence demonstrating the increased risks of cancer with age. Within our study, age was only associated with physical activity as older individuals were less likely to perform regular moderate to high intensity exercise at least 3 days a week. Older participants failing to meet these physical activity guidelines might result in higher risks for cancer compared to older adults who do meet these guidelines”

We do include results for sociodemographic influences of lifestyle related risk and protective behaviors but chose to highlight the risk behaviors performed by this sample at a higher proportion than the general population of the same age. Along with highlighting these risk behaviors (tobacco and alcohol consumption), we also provide recommendations for interventions based on sociodemographic influences.

  1. You may expand your final conclusions and future directions to up to 2 paragraphs.

Response: We thank the reviewer for these comments. Although we have added a line of content specifically to the Conclusions section 5.0 (lines 337-338), we have revised our Discussion section 4.0 to better align with our main findings and discuss future directions.

  1. Author contributions are not outlined in the required section at the end of the manuscript

Response: This has been completed through the IJERPH system and added to this document as well.

Reviewer 3 Report

In this manuscript, Ewing et al. examines the relationship between lifestyle-related cancer risk and protective behaviors among physically active young-to-middle-aged adults aged 18-49. The study was conducted through a cross-sectional, web-based survey administered to individuals who had participated in at least one recreational sporting event in the past month. The study found that men and individuals with good or excellent perceptions of health were more likely to engage in physical activity. Tobacco use was higher among men, those with less than a high-school diploma, and those preferring print-based resources over a person/provider for health information. Participants with good or excellent perceptions of health and team-based sport participants demonstrated lower tobacco use. The article suggests that an asset-based approach that leverages physical activity can reduce cancer-related risk behaviors among this unique subgroup. This study is significant for cancer prevention; however, the following issues are required for explaining:

1. The article should provide more details about the methodology and sampling techniques used in the study. This will help readers better understand the study's findings and limitations.

2. More information about the types of recreational sporting events that participants engaged in should be added.

3. The authors should provide more information about the types of cancer-related risk behaviors that were examined in the study. This will help readers understand the scope of the study and how the findings can be applied to cancer prevention efforts.

4. The authors should provide more information about the demographic characteristics of the participants, such as race/ethnicity and socioeconomic status.

5. To help readers understand how the findings can be applied in practice to reduce cancer-related risk behaviors among physically active young-to-middle-aged adults, the authors should provide more information about the implications of the study's findings for cancer prevention efforts.

6. The authors should recheck the grammar and polish the language.

Moderate editing of English language required

Author Response

  1. The article should provide more details about the methodology and sampling techniques used in the study. This will help readers better understand the study's findings and limitations.

Response: We provide additional detail of survey administration and reference to our other studies describing survey design. Refer to line# 71-72: “Further information on the survey design can be found elsewhere [14,15].”

  1. More information about the types of recreational sporting events that participants engaged in should be added.

Response: We have added detail describing the types of recreational sporting events (lines 85-89): “Open-ended text responses allowed participants to report their participation in types of recreational sports. Individual recreational sports consisted of activities such as boxing, jogging, weightlifting, cycling, martial arts, swimming, rock climbing, and yoga. Team-based sports consisted of softball, volleyball, bowling, soccer, recreational sports clubs, flag football, basketball, etc.”

  1. The authors should provide more information about the types of cancer-related risk behaviors that were examined in the study. This will help readers understand the scope of the study and how the findings can be applied to cancer prevention efforts.

Response: We have added detail describing the cancer-related risk and protective behaviors explored through this study on lines 54-55 and section 2.3:

“Complete health profiles of physically active adults could facilitate research on the combined effects of healthy (i.e., physical activity, fruit and vegetable consumption, etc.) and unhealthy (i.e., tobacco use, alcohol consumption, etc.) behaviors [6]” (Line 54-55)

  1. The authors should provide more information about the demographic characteristics of the participants, such as race/ethnicity and socioeconomic status.

Response: Please refer to the demographics subsection (3.1, line# 129) within results. Demographic results for race/ethnicity are described in lines 133-134: “Most reported non-Hispanic ethnicity (90.6%) and either White (42.4%) or Black/African American (49.2%) race”

Education, employment, income, and insurance demographics are described in line 134-138.

  1. To help readers understand how the findings can be applied in practice to reduce cancer-related risk behaviors among physically active young-to-middle-aged adults, the authors should provide more information about the implications of the study's findings for cancer prevention efforts.

Response: We thank the reviewer for these comments. We have revised our discussion section (4.0) to better align with our main findings and implications. We have also included lines# 292-295: “Based on our findings, encouraging adults to increase days of physical activity to three or more may serve as a catalyst for adopting other healthy behaviors (e.g., no tobacco use, sun protection, fruit and vegetable consumption). Future research should further examine this hypothesis.”

  1. The authors should recheck the grammar and polish the language.

Response: We have thoroughly proofread and edited our revised manuscript.

Round 2

Reviewer 3 Report

The revised manuscript has made a great improvement. I have no more comments and recommends.